# Progressive functional exercise versus best practice advice for adults aged 50 years or over after ankle fracture: protocol for a pilot randomised controlled trial in the UK - the Ankle Fracture Treatment: Enhancing Rehabilitation (AFTER) study

David J Keene [1], Matthew L Costa,[1] Elizabeth Tutton [1], Sally Hopewell,[1,2] Vicki S Barber,[1,2] Susan J Dutton,[1,2] Anthony C Redmond,[3] Keith Willett,[1] Sarah E Lamb[1,2]

For numbered affiliations see end of article.

**Correspondence to**
Dr David J Keene;
david.keene@ndorms.ox.ac.uk

## ABSTRACT

**Introduction** Ankle fractures result in significant morbidity in adults, with prognosis worsening with increasing age. Previous trials have not found evidence supporting supervised physiotherapy sessions, but these studies have not focused on older adults or tailored the exercise interventions to the complex needs of this patient group. The Ankle Fracture Treatment: Enhancing Rehabilitation study is a pilot randomised controlled trial to assess feasibility of a later definitive trial comparing best-practice advice with progressive functional exercise for adults aged 50 years and over after ankle fracture. The main objectives are to assess: (i) patient engagement with the trial, measured by the participation rate of those eligible; (ii) establish whether the interventions are acceptable to participants and therapists, assessed by intervention adherence levels, participant interviews and a therapist focus group; (iii) participant retention in the trial, measured by the proportion of participants providing outcome data at 6 months; (iv) acceptability of measuring outcomes at 3 and 6 month follow-up.

**Methods and analysis** A multicentre pilot randomised controlled trial with an embedded qualitative study. At least 48 patients aged 50 years and over with an ankle fracture requiring surgical management, or non-operative management by immobilisation for at least 4 weeks, will be recruited from a minimum of three National Health Service hospitals in the UK. Participants will be allocated 1:1 via a central web-based randomisation system to: (i) best-practice advice (one session of face-to-face self-management advice delivered by a physiotherapist and up to two optional additional sessions) or (ii) progressive functional exercise (up to six sessions of individual face-to-face physiotherapy). An embedded qualitative study will include one-to-one interviews with up to 20 participants and a therapist focus group.

**Ethics and dissemination** Hampshire B Research Ethics Committee (18/SC/0281) gave approval on 2nd July 2018.

### Strengths and limitations of this study

► The Ankle Fracture Treatment: Enhancing Rehabilitation (AFTER) study is a pilot randomised controlled trial with an embedded qualitative study recruiting at least 48 patients aged 50 years and over after an ankle fracture from at least three UK National Health Service hospitals.

► Participants will be randomly allocated to either: (i) best-practice advice (one session of face-to-face self-management advice delivered by a physiotherapist and up to two more optional advice sessions) or (ii) progressive functional exercise (up to six sessions of individual face-to-face physiotherapy).

► The interventions were developed using current research evidence and with input from clinical experts, researchers and patient and public representatives.

► We aim to assess the feasibility of a future definitive randomised controlled trial in terms of patient engagement with the trial, intervention acceptability and fidelity, retention of participants in the trial and acceptability of measuring outcomes at 3 and 6 month follow-up.

► Physiotherapists and participants are not blinded due to the nature of the interventions.

**Trial registration number** ISRCTN16612336

## INTRODUCTION

Fractures may have a devastating effect on the lower limb, resulting in mobility problems and loss of independence.[1] The incidence of ankle fractures for people aged 50 years and over in the UK is 10.4 per 10 000 person-years.[2] As the proportion of ankle fractures

occurring in older adults increases[3][4] and the population ages, a three-fold increase in these injuries by 2030 is projected.[5] A systematic review of outcome data after ankle fracture showed that functional outcomes worsen with increasing age,[6] likely due to lower physiological reserves (frailty), comorbidities, reduced muscle mass and power (sarcopenia)[7] and poor balance.[8] Participants in the Ankle Injury Management (AIM) trial (which recruited participants 60 years and over) self-reported an average 30% loss in ankle function, less confidence in walking and more fear of falling from pre-injury to 6 months post-injury.[9]

In developed countries, unstable ankle fractures typically undergo internal fixation surgery to restore normal anatomy.[10] Below-knee casts are also used in older adults as they have greater risks from surgery than younger people.[11] The AIM trial showed surgery and close contact casting for managing unstable ankle fractures in older adults had equivalence in functional outcomes.[9] Whatever the initial fracture management, protective splinting (in a cast or boot) to immobilise the ankle joint and support the injured leg is commonplace. Physical impairments after ankle fracture include ankle pain, reduced ankle motion,[12] lower limb muscle strength deficits,[13][14] mobility limitations[12][15] and walking abnormalities.[16] Many patients see a physiotherapist to aid recovery, usually in outpatient departments as most patients are ambulatory, although with difficulty.

A Cochrane review[17] of ankle fracture rehabilitation concluded that there was insufficient evidence to support traditional physiotherapy interventions targeting ankle joint and muscle impairments, such as stretching,[18] manual therapy[19] and exercise.[20] Updating the Cochrane review searches in MEDLINE and Embase, identified another multicentre trial by Moseley and colleagues.[21] They found no differences in self-reported lower limb function or quality of life between supervised exercise and a one-off advice session for adults with ankle fractures treated surgically and conservatively. Physiotherapists delivered both interventions face-to-face. The exercises included traditional ankle exercises, single-limb weight-bearing exercises and advice on using walking aids. The single advice session intervention had low adherence, with a third of participants obtaining one or more extra sessions of out-of-trial physiotherapy. Older adults were not adequately represented in either treatment group, as the mean age was 42 years.

Physiotherapy interventions tested to date have not incorporated features of advice and exercise programmes for older adults used in other rehabilitation areas. Issues such as persistent poor balance in older adults require attention.[22] Training for functional movement problems (eg, walking and stair climbing) and balance, rather than traditional ankle-impairment exercises, has a strong evidence base for older people's rehabilitation. Focusing on ankle impairments in rehabilitation may be insufficient for dealing with complex mobility problems in older adults after injury.[23]

The Ankle Fracture Treatment: Enhancing Rehabilitation (AFTER) study will assess a novel approach to physiotherapy provision, progressive functional exercise. The progressive exercise programme will use contemporary evidence-based guidelines on resistance exercise volume and load to optimise the physiological response. The widely-accepted overload principle states that movement and strength improvements require a training stimulus of sufficient volume and intensity.[24] Strong evidence suggests psychological barriers to adherence to physiotherapy advice and exercises.[25] As participants expressed fear of walking and falling in a qualitative study within the AIM trial, these psychological factors will be addressed.[26]

A pilot randomised controlled trial (RCT) is needed to investigate areas of uncertainty for a future definitive RCT, exploring trial design, recruitment, follow-up and the acceptability of the interventions and outcome measures.

## Objectives

The AFTER study will assess the feasibility of a definitive RCT to compare progressive exercise with best-practice advice in patients aged 50 years or over after ankle fracture.

The main objectives of the pilot trial are to:

▶ Assess patient engagement with the trial, measured by the participation rate of those eligible.
▶ Establish whether the interventions are acceptable to participants and therapists, assessed by intervention adherence levels, participant interviews and a therapist focus group.
▶ Determine patient retention, measured by the proportion of patients providing outcome data at 6 months
▶ Assess the acceptability of measuring outcomes at 3 and 6 month follow-up.

## METHODS AND ANALYSIS
### Study design

A multicentre pilot RCT with an embedded qualitative study. Participants will be allocated to either: (i) best-practice advice (one session of face-to-face advice delivered by a physiotherapist, with up to two further optional advice sessions) or (ii) progressive functional exercise (up to six sessions of individual face-to-face physiotherapy) (see figure 1).

### Setting

Recruitment will be from at least three National Health Service (NHS) hospitals and their related physiotherapy services. Participants will be identified during inpatient stays or when attending fracture clinics, where they will be screened and given study information. Eligibility will be assessed by the treating clinician, usually an orthopaedic surgeon or physiotherapist.

### Study participants

The target population is at least 48 adults aged 50 years or over attending NHS services to manage ankle fractures

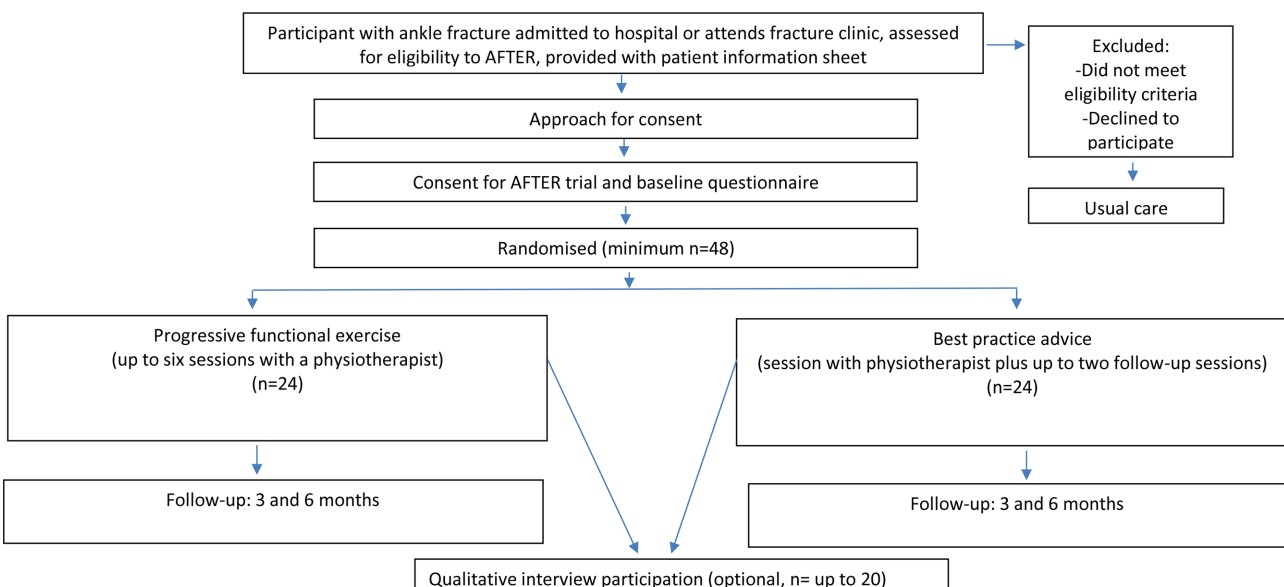

**Figure 1** Study flow diagram for the Ankle Fracture Treatment: EnhancingRehabilitation (AFTER) study.

that require definitive management with surgical treatment or ankle immobilisation for at least 4 weeks.

## Eligibility

We will include adults aged 50 years or over with an ankle fracture who are undergoing surgical fixation or conservative management involving ankle immobilisation for at least 4 weeks.

Patients will be excluded if they:

► are unable to adhere to trial procedures or complete questionnaires
► do not have capacity to consent to study participation
► were not ambulatory before the injury
► are considered inappropriate for referral to physiotherapy by the clinician
► cannot attend outpatient physiotherapy at a participating centre
► have serious concomitant disease (such as terminal illness)
► have bilateral lower limb fractures
► have an ipsilateral concurrent pilon fracture
► have open fracture wounds, external fixation or substantial skin loss or grafts that would limit ankle or lower leg exercise

## Recruitment

Participants will be recruited from NHS hospitals and related physiotherapy services. Posters displayed in clinics will advertise the AFTER study to patients and clinicians.

## Screening and eligibility assessment

Potential participants will be inpatients or attending trauma and orthopaedic clinics. Patients presenting with an ankle fracture will be assessed against the eligibility criteria, given a copy of the participant information sheet and asked if they wish to be considered for the study. Patients who meet the eligibility criteria and would like to participate will be approached for informed consent. If

the potential participant is deemed eligible and is willing to proceed, then they and the researcher will sign and date a consent form.

Patients who do not meet the eligibility criteria or who do not wish to participate will receive standard NHS treatment. We will record the age and gender of those not eligible or who decline participation to assess the generalisability of those recruited. We will ask these patients why they declined the study and record any answers provided.

## Randomisation

Consented participants will be randomised 1:1 to the intervention groups using the centralised computer randomisation service provided by the Oxford Clinical Trials Research Unit. The recruiting site's research facilitator will undertake randomisation directly themselves or will contact the study office over the telephone to access the system on their behalf. Randomisation will be computer-generated and stratified by centre and initial fracture management (surgical or non-surgical) using a variable block size to ensure participants from each centre have an equal chance of receiving each intervention. Participants will only be randomised after eligibility assessment and informed consent.

## Blinding

Physiotherapists delivering the intervention and study participants will be told the treatment allocation. Researchers independent of the clinical team will collect the objective outcome measures at 6 month follow-up.

## Interventions

### Training and monitoring of intervention delivery

The best-practice advice and progressive functional exercise interventions will be provided face-to-face and one-to-one by physiotherapists. The study team will give the treating therapists training and an intervention manual on the theory and practical delivery of the interventions.

Treating physiotherapists will record the delivery and content of the sessions attended by each participant.

Patient pathways vary at different hospitals so patients may have had instruction on basic ankle exercises, wound care and use of walking aids from a physiotherapist or other health professional while an inpatient or in an acute fracture clinic before starting the study interventions. This will be recorded in the study treatment log. The intervention session(s) can be tailored to the patient's recovery level, as per usual care. Some centres make referrals for physiotherapy while patients are inpatients or at earlier clinic follow-ups. Other centres delay referrals until a clinic review several weeks later. The study will adapt screening and recruitment to embed in the patient pathway as much as is practical. The process at each centre will be reviewed as the study progresses.

The best-practice advice or progressive functional exercise sessions will be given when the participant can mobilise with unrestricted weight-bearing and do ankle exercises as guided by their surgeon or physiotherapist. We anticipate that this will usually be around 6 to 8 weeks post-injury.

### Best-practice advice

A single advice session with a physiotherapist with one or two follow-up appointments is commonplace in the UK NHS. This was considered an acceptable provision model reflecting best-practice advice in a consensus meeting with healthcare professionals and patient and public involvement representatives held to help design the AFTER study. There are no clinical guidelines on physiotherapy for ankle fractures. Recent National Institute for Health and Care Excellence (NICE) fracture guidelines made no specific recommendations for ankle fracture physiotherapy, but did highlight the importance of supporting advice with written information to enhance self-management.[27]

There is variation in physiotherapy provision in the NHS. However, as with many physiotherapy trials, the volume and content will be standardised to manage heterogeneity and ensure a reproducible intervention that can be assessed for generalisability. This intervention aims to be a credible representation of current best-practice advice across the NHS. Key stakeholders, including clinical experts, researchers and patient and public representatives, contributed substantially to the development of the interventions during the design phase of the AFTER study.

The best-practice advice intervention will focus on self-management. It will include a 20 to 60 min session (depending on local service provision) of assessment, education, reassurance and detailed self-management advice on ankle exercises, gait training, stair climbing, walking aid advice and basic balance exercises. An advice booklet will provide key information.

Up to two further sessions advice will be optional for participants that are having difficulties with self-management or the exercises. The physiotherapist's role

will be to reassess and re-enforce self-management advice. During the initial stage of recruitment these additional sessions may involve a telephone call or an additional face-to-face contact. In the latter stage of recruitment, these sessions will be offered as telephone consultations only. Use of additional sessions will be recorded and monitored.

Physiotherapists use a range of passive manual and electrotherapy modalities but clinical trials and systematic reviews have found limited evidence that these treatments effectively improve outcomes after ankle fracture.[17] They will therefore not be a core part of the best-practice advice but use will be recorded in the treatment logs.

### Progressive functional exercise

Participants will undertake progressive functional resistance training and balance exercises. A physiotherapist will instruct on, supervise and progress the exercises in up to six sessions over 16 weeks. This period allows sufficient time for neuromuscular adaptation to exercise.[28] The programme can end early if all rehabilitation goals are achieved in under six sessions. The first session will be 20 to 60 min and the rest up to 30 min, consistent with physiotherapy sessions in the NHS. Physiotherapists will provide assessment, advice, education about progressing recovery, gait training, walking aid instruction and an information booklet.

The programme will be highly structured but calibrated for each individual. Tailoring programmes to the participant is a key feature of effective interventions in older adults.[29] All participants will receive a core set of functional lower limb strengthening exercises in line with the evidence for improving muscular strength and power in older persons' rehabilitation.[30] Strength improvements do not necessarily translate to function and balance improvements. The programme will include supervised gait training to target major walking difficulties after ankle fracture. Gait training improves motor control during walking in older adults, which is related to functional mobility and balance improvements.[31] Balance exercises will also be included in the programme and introduced once the participant is able to weight bear sufficiently to perform these. Exercises will be practised in the clinic but conducted at home to achieve an effective dose. Based on the participant's functional goals, exercises will be progressed to make them task-specific, for example walking on uneven surfaces or slopes, climbing stairs or jogging. The participants will receive a personal exercise guide and diary. Exercise progression will be based on evidence-based guidelines[24] but individualised by progressing and regressing the volume and load in line with each participant's capabilities and preferences.

The progressive functional exercise intervention will use simple health behaviour change techniques to optimise adherence to home exercise. We drew on the evidence-based NHS Health Trainers Handbook, recommended for routine use by health professionals to promote patient behaviour change.[32] Our group has used

these techniques in other physiotherapy trials to develop feasible exercise programmes that have resulted in self-efficacy improvements.[33] The strategies use a two-stage mechanism, increasing intention to adhere to the exercise regimen and translating this behavioural intention into actual behaviour.

Participants will be asked to identify their goals following usual physiotherapy practice and, with the treating therapist's help, write an action plan for where and when they will perform their home exercises and a contingency plan for managing difficulties.

Therapists will be trained to focus on helping participants identify barriers to exercise and becoming more physically active post-injury, and facilitating problem-solving. The therapists will offer education on how exercise and physical activity can help participants to achieve their goals and will reassure participants about their capacity to exercise and increase their physical activity.[34] The intervention will give participants individualised feedback on their rehabilitation progress and reinforcement over the sessions, and will facilitate identification of barriers to doing the home exercise programme, which all have a strong evidence base to support their use in older adults.[29]

There is evidence that patients do not retain much of the information received face-to-face.[35] This sense of being overwhelmed by verbal instructions from physiotherapists was echoed by several patient and public involvement (PPI) group representatives, who strongly advocated quality supporting materials. A high-quality information booklet will be developed by the AFTER study team and provided to participants.

### Concomitant care

Other aspects of health and social care will continue as normal. Analgesia use will be self-reported. Participants may seek other forms of treatment during follow-up, but will be asked to use their usual routes of access or referral to do so. Additional treatments, including contact with their general practitioner or other health professionals, will be recorded in participant follow-up questionnaires.

A rigorous quality control programme will ensure intervention fidelity.[36] Participants can seek care outside the study, which will be recorded as part of health resource use. Participant crossover between intervention groups will not be allowed. The local site coordinating physiotherapist and study team will share responsibility for intervention quality control. Site visits will be conducted periodically to observe recruitment, consent, randomisation, data collection and progressive exercise and best-practice advice sessions. Permission will be sought from the participants to observe treatment sessions. Data will be collected on intervention delivery, number of treatment sessions attended and details about the core and adaptable components to facilitate monitoring and reporting. Sites will regularly receive feedback from quality control visits to help maintain and improve fidelity. Identified issues will be addressed by engaging the site staff in more training and increasing monitoring by the central trial team.

### Outcome measures

#### Feasibility criteria

The main aim of this pilot RCT is to determine the feasibility of a future definitive trial.[37] The focus will not be on a primary outcome of effectiveness, but instead on meeting success criteria. The main uncertainty is whether patients find it acceptable to be randomised to different types of physiotherapy. Screening data from the AIM trial gives confidence that there are enough potential participants to investigate the feasibility criteria. To determine the feasibility of a definitive RCT, the success criteria are:

▶ A study participation rate of at least 25% of those eligible, to indicate acceptability and generalisability.
▶ At least 48 eligible participants across at least three sites agree to participate over a maximum of 18 months.
▶ At least 85% of participants complete the study intervention sessions.
▶ At least 80% of participants attend study follow-up at 6 months.

Descriptions from the participant interviews and therapist focus group that indicate randomisation/interventions are acceptable will also be used to assess feasibility. Treatment logs will monitor intervention fidelity and tolerability. Adherence to home exercise will be monitored via participant self-reports. An estimate of SD for the primary outcome is not required as the AIM trial will provide this information for the definitive trial.

#### Outcomes

Outcomes will be collected to assess the feasibility of their collection in a future definitive RCT (see tables 1 and 2).

Patient-reported outcomes at 3 and 6 months will be:

▶ ankle-related symptoms and function: Olerud and Molander Ankle Score[38]
▶ lower-limb function limitations: Lower Extremity Functional Scale[39]
▶ pain: Visual Analogue Scale, 0 to 100 scale health-related quality of life: EQ-5D-5L score[40]
▶ fear of falls: Falls Efficacy Scale-International (short version)[41]
▶ self-efficacy: self-efficacy exercise score[42]
▶ return to desired activities, including work, social life and sport activities
▶ walking aid use and distances
▶ exercise adherence

At 6 month follow-up, a blinded outcome assessor will collect objective measures of ankle function and physical performance:

▶ ankle joint range: hand-held goniometry[43]
▶ muscle strength: hand-held dynamometry (Lafayette Manual Muscle Test System, Lafayette Instrument, Indiana, USA) of ankle dorsi/plantarflexion using a 'make' approach (working up to a maximal contraction over a maximum of 5 s and without pushing into

**Table 1** Time points at which the outcomes will be assessed

| Time point | Enrolment | Allocation | 0–4 months* | 3 month follow-up* | 6 month follow-up |
|---|---|---|---|---|---|
| Screening log | ✓ | | | | |
| Informed consent | ✓ | | | | |
| Eligibility confirmed | ✓ | | | | |
| Randomisation | | ✓ | | | |
| Control: best practice advice | | | one physio session (up to two more if struggling) | | |
| Intervention: progressive functional exercise | | | Up to six physio sessions | | |
| Baseline questionnaire | ✓ | | | | |
| Follow-up questionnaire | | | | ✓ | ✓ |
| Follow-up clinic visit at hospital | | | | | ✓ |
| Follow-up reminders | | | | ✓ | ✓ |
| Qualitative interview (optional) | | | ✓ | | |

*The first 3 month follow-up occurs while participants are doing the recommended exercises, and for the progressive exercise group, they may still be returning for sessions with the physiotherapist.

pain and the assessor maintaining position of the device).[44] Participants will be measured three times and have at least 10 s rest between attempts. Self-reported body weight will be recorded to aid interpretation of strength measures.

► mobility and balance: short physical performance battery.[45] The test involves physical tests of balance, walking speed and repeated rises from a chair. It has been used extensively in clinical trials owing to its practical utility, the strong evidence base for its measurement properties, and its relationship to frailty, risk of falls and disability in older adults.[46]

Data on health resource use (consultation with primary and secondary care, prescribed and over-the-counter medication use, additional physiotherapy and hospital admission), additional out-of-pocket expenses and work absence (number of sickness days) will also be collected to inform a future definitive RCT. A full health economic evaluation will not be conducted. Data will be collected alongside the other outcome measures in participant questionnaires.

### Adverse events

Foreseeable adverse events (AE) occurring as a result of the trial intervention(s) will be recorded. Participants will receive information on the potential AEs resulting from the treatment exercises and what they should do if they experience an AE, as would happen as part of standard NHS procedures.

Expected general side effects of any exercise, such as delayed-onset muscle soreness and temporary increases in pain <1 week, will not be recorded as AEs. Pain increases >1 week will be recorded in patient-reported questionnaires. Although unlikely, any exacerbations of

other medical conditions during exercise or exercise-related injurious falls will be recorded in patient-reported questionnaires or by the site investigators if they become aware of such an event.

A serious AE (SAE) is any unexpected untoward medical occurrence related to the trial interventions that results in death, is life-threatening, requires inpatient hospitalisation or prolongation of existing hospitalisation, or results in persistent or significant disability/incapacity. SAEs are likely to be rare and are unlikely to occur as a result of the exercise programmes delivered in this study. If an SAE arises between study enrolment and final follow-up visit and is deemed related to the trial interventions, the Clinical Trials Unit standard operating procedures will apply.

### Follow-up data collection

Participants will be followed-up 3 months after randomisation with a postal questionnaire and 6 months after randomisation face-to-face at the hospital. They will be offered telephone, postal or electronic follow-up if they are unable to attend the 6 month follow-up.

Participants will be asked to complete the 3 month (and, if applicable, 6 month) questionnaire(s) and return it to the AFTER trial team. Those who do not respond to the initial questionnaire will be sent at least one reminder by telephone, SMS text messaging or email. Telephone and electronic follow-up will also be used to collect a core set of questionnaire items if these are not completed on the returned questionnaire.

### Sample size

The main feasibility objective and therefore the basis of the sample size estimate is participant recruitment at three centres with a staggered start. The target sample size

**Table 2** Participant timeline

| Outcome | Measurement | Time point |
|---|---|---|
| Demographic | age, gender, height, weight, ethnicity, smoking, alcohol consumption, date of injury, fracture classification, initial fracture management, pre-injury walking aid use and exercise tolerance, current work status, level of education, place of residence, social support | Baseline |
| Ankle-related symptoms and function | Olerud and Molander Ankle Score[38] | Baseline, 3 and 6 months |
| Lower limb function limitations | Lower Extremity Functional Scale | Baseline, 3 and 6 months |
| Pain | Visual Analogue Scale | Baseline, 3 and 6 months |
| Health-related quality of life | EQ-5D-5L score[40] | Baseline, 3 and 6 months |
| Fear of falls | Falls Efficacy Scale-International (short) | Baseline, 3 and 6 months |
| Self-efficacy | Self-efficacy exercise score[42] | 3 and 6 months |
| Return to desired activities, including work, social life and sport activities | Patient-reported return to activities | 3 and 6 months |
| Walking aid use and distance | Patient-reported use of aids and maximum distance walked on any single occasion | Baseline (recall pre-injury and current status), 3 and 6 months |
| Adherence to exercise | Patient-reported exercise performance | 3 and 6 months |
| Mobility and balance | Short physical performance battery[45] | 6 months |
| Ankle joint range | Hand-held goniometry[43] | 6 months |
| Muscle strength | Hand-held dynamometry | 6 months |
| Medication usage | Prescribed and over-the-counter medications | Baseline, 3 and 6 months |
| Work disability | Sick leave (days) | Baseline, 3 and 6 months |
| Healthcare use | NHS outpatient and community services (eg, general practitioner, additional physical therapy) NHS inpatient and day case (eg, radiography, readmissions) Private healthcare services | 3 and 6 months |
| Out-of-pocket expenses | Patient-related out-of-pocket expenses recording form | 3 and 6 months |
| Adverse events | Patient-reported adverse events | 3 and 6 months |

NHS, National Health Service.

is a minimum of 48 participants. Based on three sites and staggered starts, this is equivalent to at least 1.5 participants per month per site over 12 months. This sample size will enable an estimate of a minimum 25% recruitment rate of those eligible to within a 95% CI of ±6% (calculated using the modified Wald method).[47] If the sample size is achieved sooner than 12 months, recruitment will continue up to a maximum of 60 participants to enable further feasibility assessment regarding the change to the best practice advice intervention. Recruitment will cease if the target minimum sample size is not achieved within 18 months. We have selected a conservative recruitment rate as recruitment to previous ankle fracture exercise trials in younger adults with less complex health needs has been 37% to 80%.[20 21]

### Statistical analysis

Feasibility outcomes will be reported, including the number of participants who are approached, are eligible, consent to randomisation and are followed-up, attendance of intervention sessions, completion rates of exercise diaries and data completeness. Baseline characteristics will be reported using descriptive statistics, per group and overall, using mean and SD (or median and IQR if non-normally distributed), and minimum and maximum, for continuous variables and number and percentage of patients in each group for binary or categorical variables.

Clinical outcome measures will be reported descriptively. Differences between treatments for the intention-to-treat population will be reported with 95% CIs. Withdrawals from treatment and the trial will be reported, with reasons where provided. AEs, and SAEs will be reported, both the number of participants that experience an event and the total number.

The primary statistical analysis will be carried out on the basis of intention-to-treat, with all randomised participants included and analysed according to their allocated treatment group, irrespective of which treatment they actually receive or their treatment compliance.

### Embedded qualitative study

Investigating patient experience is fundamental to understanding how interventions affect patients' lives and will give insights into the acceptability of the trial interventions and randomisation to them. The embedded qualitative study aims to find out more about the patients' experience of the two interventions within the context of their recovery from ankle fracture. This understanding will help us review the acceptability of the two interventions, which aspects help or hinder recovery. This will enable us to refine the interventions, retaining aspects that are important to patients and developing or removing those that are less helpful. For instance, patients may struggle with the pace or complexity of the progressive exercises, which could then be modified for the future definitive trial. In addition the interviews will provide valuable insight into how patients experience the trial processes.

For example we know that patients take part in studies because they place their trust in the clinical team, they can have trouble understanding some information, can find randomisation unacceptable in clinical situations and have therapeutic misconceptions.[48–50] Gaining an insight into the aspects that facilitate and limit participation in the study will help us refine our information processes to include areas that are of concern to patients.

Interviews will be undertaken with a purposive sample of up to 20 participants from recruiting centres and each intervention group, approximately 4 months after randomisation. Participants will be invited to take part and provide their agreement to be contacted for an interview after they have consented to the main pilot trial. Written informed consent will be provided prior to the interview. The interviews will provide an insight into patient experience of being in the trial, the interventions and recovery and outcomes that are important to them within the context of their life. Factors that inhibit or facilitate their ability to fully take part in the study will be used to inform a large-scale definitive RCT.

To sensitise the research team to factors that are important to study participants the semi-structured interview schedule will be informed by input from a PPI member and a therapist. Open questions will be used to ensure participants can talk freely about what is important to them. Participants will choose whether to be interviewed in the local hospital, over the telephone or at home.

The experiences of therapists delivering the trial interventions and recruiting patients will also be explored. Therapists will be invited to participate in a focus group to be held at the end of the recruitment phase.

Interviews and the focus group will be audio-recorded, transcribed verbatim and NVivo (QSR International Pty Ltd, Melbourne, Australia) will be used to help manage the data. Data will be coded and grouped into categories and themes drawing on thematic analysis.[51] Any quotes used in reporting of the findings will be anonymised.

### Patient and public involvement

The development of the study funding application, intervention development and study materials were supported by a patient and public involvement group, who will also be involved in developing the dissemination strategy.

### ETHICS AND DISSEMINATION

The AFTER study has been prospectively registered. All protocol amendments will be subject to review by the Sponsor (Oxford University), the ethics committee and Health Research Authority and will be included in the final report. All data will be processed following Oxford Clinical Trials Research Unit standard operating procedures. Adverse events and serious adverse events are likely to be rare. If they occur they will be reviewed by the trial management group.

This protocol has been reported following the Standard Protocol Items: Recommendations for Interventional Trials (SPIRIT) statement.[52] Results will be published in an open-access journal, reported following the Consolidated Standards of Reporting Trials (CONSORT) guidelines for pilot and feasibility trials.[53 54] The Template for Intervention Description and Replication (TIDieR) statement will be used to report the intervention,[55] ensuring replication is possible. Results will be published in a peer-reviewed journal with authorship eligibility according to the International Commitee of Medical Journal Editors (ICMJE) criteria. Participants will be asked if and how they would like to be informed of the study results during the consent process. We will share study results before publication with those participants who request it.

**Author affiliations**
[1]Nuffield Department of Orthopaedics, Rheumatology and Musculoskeletal Sciences, University of Oxford, Oxford, UK
[2]Centre for Statistics in Medicine, University of Oxford, Oxford, UK
[3]Leeds Institute of Rheumatology and Musculoskeletal Medicine, University of Leeds, Leeds, UK

**Acknowledgements** The AFTER study collaborating Principal Investigators are: Hannah Perkins (Oxford University Hospitals NHS Foundation Trust), Carol McCrum (East Sussex Healthcare NHS Trust), Jacky Jones (Guy's and St Thomas' NHS Foundation Trust), Carey McClellan (University Hospitals NHS Foundation Trust), Fiona Cowell (Royal Liverpool and Broadgreen University Hospital NHS Trust). We acknowledge English language editing by Dr Jennifer A de Beyer of the Centre for Statistics in Medicine (University of Oxford), and input during the intervention development phase by Colin Forde (Oxford University Hospitals NHS Foundation Trust) as part of an NIHR MRes research internship.

**Contributors** DJK is the chief investigator and conceived and designed the study, was awarded the funding and drafted the manuscript. MLC, ET, SH, VSB, SJD, ACR, KW and SEL contributed to study design and provided specific content and edited the manuscript. SJD oversees the statistical aspects of the study. ET oversees the qualitative substudy. VSB has provided trial management oversight. All authors have reviewed and approved the manuscript.

**Funding** This report is independent research supported by the National Institute for Health Research (NIHR Post-Doctoral Fellowship, Dr David Keene, PDF-2016-09-056). The report was supported by the NIHR Biomedical Research Centre, Oxford. Professor Lamb receives funding from the NIHR Collaboration for Leadership in Applied Health Research and Care Oxford at Oxford Health NHS Foundation Trust.

**Disclaimer** The views expressed in this publication are those of the authors and not necessarily those of the NHS, the National Institute for Health Research or the Department of Health and Social Care. The study sponsor and funders had no role in study design, writing of the report, and the decision to submit the report for publication.

**Competing interests** None declared.

**Patient consent for publication** Not required.

**Ethics approval** The study has been approved by the South Central – Hampshire B Research Ethics Committee (Ref: 18/SC/0281) and the Health Research Authority.

**Provenance and peer review** Not commissioned; externally peer reviewed.

**ORCID iDs**
David J Keene http://orcid.org/0000-0001-7249-6496
Elizabeth Tutton http://orcid.org/0000-0003-3973-360X

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
