## [Reviewer comments · BMJ Open]

ARTICLE DETAILS

TITLE (PROVISIONAL)	Progressive functional exercise versus best practice advice for adults aged 50 years or over after ankle fracture: protocol for a pilot randomised controlled trial in the United Kingdom - the Ankle Fracture: Enhancing Rehabilitation (AFTER) study
AUTHORS	Keene, David; Costa, Matthew; Tutton, Elizabeth; Hopewell, Sally; Barber, Vicki; Dutton, Susan; Redmond, Anthony; Willett, Keith; Lamb, Sarah

VERSION 1 – REVIEW

REVIEWER	Dana Loudovici-Krug University Hospital Jena, Institute of Physiotherapy, Germany
REVIEW RETURNED	08-May-2019

GENERAL COMMENTS	It is a detailed study protocol, the trial process is comprehensible.
---

REVIEWER	Sherif Mohamed Abdelgaid Helwan University, Cairo, Egypt
REVIEW RETURNED	09-Jun-2019

GENERAL COMMENTS	The protocol is nicely presented both in its overview of the subject and the reporting study design. It is written in a clear scientific style and to the point. However, the proposal is too long, please can you reduce it by taking out non-essential words and repeated phrases. Also, there is a lack of updated references. (in the last 5 years: 8 references in 2015 & 2016/ 1 reference in 2017/ 0 references in 2018 & 2019). In recent publications, you may find is a "right answer," to your research question.
--

REVIEWER	Aryelly Rodriguez University of Edinburgh, UK
REVIEW RETURNED	17-Jul-2019

GENERAL COMMENTS	This is a very well thought out study, I just have some minor comments: 1.-I would choose One (or maximum two) primary objective (which in turn would link to one (two) primary outcome(s)). 2.-For the abstract it is said: "The main objectives are to assess: (i) patient engagement with the trial, measured by the participation rate;..." and also in objectives: "Assess patient engagement with the trial, measured by the participation rate". Please clarify this is the participation rate at eligibility, as this would make it more specific and link it to the "Feasibility criteria " which states: "A study participation rate of at least 25% of those eligible, to indicate acceptability and generalisability"
	3.- In the following statement: "If an SAE arises between study enrolment and final follow-up visit and is deemed related to the trial interventions, standard operating procedures will apply.", which SOPs? please add a reference 4.-Under Sample size, please clarify how 48 was obtained, because $(1.5 \text{ participants}) \times (3 \text{ sites}) \times (12 \text{ months}) = 54$. Please, also clarify that you would need to approach 192 eligible patients in order to recruit 48. Finally, I am struggling to match this up with the "Feasibility Criteria" which seems less flexible requiring "A study participation rate of at least 25% of those eligible" and "At least 48 eligible participants across at least three sites...", 48 is the centre of the interval, so if you were to recruit lets say 45 patients (23%) that is within the 95% CI for your estimated proportion (18.9, 31.1), but you would have failed the criteria as $45 < 48$ and $23 < 25$. Under these circumstances, I think it is more logical to set 48 as the lower limit for the 95% CI (i.e. you would need to approach more patients). I hope that make sense. Usually for pilot/feasibility studies it is not required to do a sample size calculation, because very little information is available, I am gladly surprise that you have put it together. 5.- Please for "Feasibility outcomes will be reported, including the number of participants who are approached, are eligible, ... all with 95% confidence intervals." clarify that those parameter are denominators and a 95% CI would not be calculated, unless there is a bigger set of patients from which they are going to come from, if so, please state it. 6.- Under "using mean and standard deviation (or median and interquartile range if non-normally distributed)" , if possible add minimum and maximum. 7.-Please under "Withdrawals from treatment and the trial, AEs, and SAEs will be reported. ", please clarify, for withdrawals if you are planning to report only numbers or numbers and reasons, the same for AE and SAE (which are usually presented with number of patients with at least one AE and/or SAE, and total number of AEs and SAEs)
REVIEWER	Guangyu Tong Department of Psychiatry and Behavioral Sciences, Duke University
REVIEW RETURNED	25-Jul-2019

GENERAL COMMENTS	The protocol under review is aimed to study the potential differences in the recovery of ankle fracture between two treatments (progressive functional exercise vs. best practice advice). While the trial design is comprehensively explained in this protocol, I have the following eight comments that may help the authors clarify and improve their study. 1. Definition of the treatments: regarding the second point on p4, the “up to X sessions” for each treatment seems to indicate some within-treatment heterogeneity. I am concerned such heterogeneity would affect the observed outcomes. Does the “up to X sessions” also reflect the willingness to treatment, which incurs self-selection bias ? Some clarifications are needed here. 2. The sample size issue: while on p.15 the authors acknowledge the recruitment rate is related to the sample size, which is 24 for each group, the authors did not justify the statistical power with this sample size. It is important to show through power calculation that the designed sample size can give your analysis sufficient power.
---

	3. Clustering issue: the authors mentioned on p.2, line 23-29 “48 participants... will be recruited from three national health service hospitals...”. Therefore, it is helpful to clarify the following questions: Does it mean there will be clusters in the recruitment? Does this need to be addressed in the statistical analysis? How is randomization conducted within/between hospitals? More details/clarifications are necessary. 4. Mediating effect of psychological barrier: on p.6. 28-32, the authors mentioned that “psychological barriers to adherence to physiological advice and exercises... will be addressed”. Can the authors provide more details on how this issue could be addressed in your trial? Do people under the two treatments experience different psychological barriers in average? How would this be addressed/balanced in the design/analysis? 5. The statistical analysis section on p.15-16 needs to be improved. The current analytical plan seems to only calculate the group means and standard deviations. Is there any power calculation for this analysis? Should more information/covariates be included in the analysis since there are only 24 participants in each arm? Personal thoughts: at least, some additional variate adjustment analysis is necessary to improve the power. Also, because the participants are not blinded and the non-compliance rate might be high, the ITT (intention to treatment) estimand (true effect size) could naturally be very small. It is therefore even more important to show whether 24 people in each treatment arm can provide enough power to your analysis. 6. Missing data: it is likely that the current trial will encounter missing data/dropout issues. It is discussed in the feasibility section (p.12-13). However, how the missing data would be addressed in the analysis, such as by using weighting- or imputation-based methods, is not discussed. The validity of the analysis is likely to be based on the “missing at random” assumption. And if there is any informative dropout, such as psychological barrier/lack of efficacy, how can this information be incorporated into the analysis? Some more discussion is needed for this. 7. Embedded qualitative study: it is not particularly clear to me whether the embedded qualitative study is directly related to this trial design. While the authors state on p.16, “...will give insights into the acceptability of the trial interventions and randomization to them,” the analysis of this trial will be based on ITT (intention to treatment), which means the data collected in this embedded qualitative study will not affect the conclusion of the study. The authors need to either provide a stronger justification over this part or remove it. 8. Minor point: I am confused about the column names in Table 1, on p.24, why are the sessions appeared in 0-4 months, but the first follow up is 3-month? So, it is possible for data collection/follow-up to happen even before the treatment sessions are finished? Some clarifications are needed here.
--	--

VERSION 1 – AUTHOR RESPONSE

Reviewer: 1

Reviewer Name: Dana Loudovici-Krug

Institution and Country: University Hospital Jena, Institute of Physiotherapy, Germany

Please state any competing interests or state 'None declared': None declared

Please leave your comments for the authors below

It is a detailed study protocol, the trial process is comprehensible.

RESPONSE: Thank you for this positive feedback.

Reviewer: 2

Reviewer Name: Sherif Mohamed Abdelgaid

Institution and Country: Helwan University, Cairo, Egypt

Please state any competing interests or state 'None declared': None declared

Please leave your comments for the authors below

The protocol is nicely presented both in its overview of the subject and the reporting study design. It is written in a clear scientific style and to the point.

However, the proposal is too long, please can you reduce it by taking out non-essential words and repeated phrases.

RESPONSE: In the process of revision we have been mindful of this feedback, aiming to make some of the changes while not exceeding the 4000 word maximum.

Also, there is a lack of updated references. (in the last 5 years: 8 references in 2015 & 2016/ 1 reference in 2017/ 0 references in 2018 & 2019). In recent publications, you may find is a "right answer," to your research question.

RESPONSE: Prior to submission we updated searches and have repeated this for the revision and there are no newer trials assessing physiotherapy after ankle fracture.

Reviewer: 3

Reviewer Name: Aryelly Rodriguez

Institution and Country: University of Edinburgh, UK

Please state any competing interests or state 'None declared': None

Please leave your comments for the authors below

This is a very well thought out study, I just have some minor comments:

1.-I would choose One (or maximum two) primary objective (which in turn would link to one (two) primary outcome(s)).

RESPONSE: In line with current recommendations for the design of feasibility studies, we have not selected a single primary objective and outcome (Eldridge et al 2016, BMJ; 355). Instead we have a range of success criteria related to the main feasibility study objectives. The aim of the feasibility study is not to test a specific question/hypothesis, but rather to assess a range of design uncertainties for a future definitive trial, where a primary outcome and endpoint would be required.

2.-For the abstract it is said: "The main objectives are to assess: (i) patient engagement with the trial, measured by the participation rate;..." and also in objectives: "Assess patient engagement with the trial, measured by the participation rate". Please clarify this is the participation rate at eligibility, as this would make it more specific and link it to the "Feasibility criteria " which states: "A study participation rate of at least 25% of those eligible, to indicate acceptability and generalisability"

RESPONSE: Amended as suggested.

3.- In the following statement: "If an SAE arises between study enrolment and final follow-up visit and is deemed related to the trial interventions, standard operating procedures will apply.", which SOPs? please add a reference

RESPONSE: We added that these are the SOPs of the Clinical Trials Unit.

4.-Under Sample size, please clarify how 48 was obtained, because $(1.5 \text{ participants}) \times (3 \text{ sites}) \times (12 \text{ months}) = 54$. Please, also clarify that you would need to approach 192 eligible patients in order to recruit 48. Finally, I am struggling to match this up with the "Feasibility Criteria" which seems less flexible requiring "A study participation rate of at least 25% of those eligible" and "At least 48 eligible participants across at least three sites...", 48 is the centre of the interval, so if you were to recruit lets say 45 patients (23%) that is within the 95% CI for your estimated proportion (18.9, 31.1), but you would have failed the criteria as $45 < 48$ and $23 < 25$. Under these circumstances, I think it is more logical to set 48 as the lower limit for the 95% CI (i.e. you would need to approach more patients). I hope that make sense. Usually for pilot/feasibility studies it is not required to do a sample size calculation, because very little information is available, I am gladly surprise that you have put it together.

RESPONSE: The sample size calculation of 48 over 12 months at 1.5 participants per month was based on a staggered start at the sites. It is because of the staggered start that the calculation proposed above does not correlate. The staggered start is mentioned twice in the sample size section so no further details have been added at this stage. We estimated 1.5 participants would be a realistic target based on our experience in conducting trauma trials in this patient group. It would mean a later definitive trial could be conducted within the practical and financial constraints of a full scale trial.

We are uncertain as to how many eligible participants will be screened but we recognised that if a very low proportion of those eligible agree to participate that it would indicate a definitive trial may be unfeasible. The sample size of 48 enables an estimate of 25% recruitment rate of those eligible, with a 95% CI of 19 to 31%.

To clarify this further we have amended the sample size section to clarify that the 25% recruitment is of those eligible.

5.- Please for "Feasibility outcomes will be reported, including the number of participants who are approached, are eligible, ... all with 95% confidence intervals." clarify that those parameter are denominators and a 95% CI would not be calculated, unless there is a bigger set of patients from which they are going to come from, if so, please state it.

RESPONSE: Thank you for this feedback, we have removed the 95% CI calculation from this section.

6.- Under "using mean and standard deviation (or median and interquartile range if non-normally distributed)", if possible add minimum and maximum.

RESPONSE: We added minimum and maximum, we agree this will be a useful addition.

7.-Please under "Withdrawals from treatment and the trial, AEs, and SAEs will be reported. ", please clarify, for withdrawals if you are planning to report only numbers or numbers and reasons, the same for AE and SAE (which are usually presented with number of patients with at least one AE and/or SAE, and total number of AEs and SAEs)

RESPONSE: We have added the detail as follows:

'Withdrawals from treatment and the trial will be reported, with reasons where provided. AEs, and SAEs will be reported, both the number of participants that experience an event and the total number.'

Reviewer: 4

Reviewer Name: Guangyu Tong

Institution and Country: Department of Psychiatry and Behavioral Sciences, Duke University

Please state any competing interests or state 'None declared': None declared

Please leave your comments for the authors below

The protocol under review is aimed to study the potential differences in the recovery of ankle fracture between two treatments (progressive functional exercise vs. best practice advice). While the trial design is comprehensively explained in this protocol, I have the following eight comments that may help the authors clarify and improve their study.

1. Definition of the treatments: regarding the second point on p4, the "up to X sessions" for each treatment seems to indicate some within-treatment heterogeneity. I am concerned such heterogeneity would affect the observed outcomes. Does the "up to X sessions" also reflect the willingness to treatment, which incurs self-selection bias ? Some clarifications are needed here.

RESPONSE: The interventions are designed to be pragmatic and reflect clinical practice in the UK. The best practice advice intervention is designed to be a single session intervention focussing on selfmanagement, with additional sessions available only for people that are having difficulties with selfmanagement. The progressive exercise intervention is designed to be up to 6 session, discontinuing before the maximum if rehabilitation goals are met. It would be irregular in the UK to provide additional sessions after all rehabilitation goals are met, so we do not stipulate the full 6 sessions need to be provided. We are aware that heterogeneity could be a problem and this is why adherence to the interventions per protocol is being carefully monitored and reported in this feasibility study.

2. The sample size issue: while on p.15 the authors acknowledge the recruitment rate is related to the sample size, which is 24 for each group, the authors did not justify the statistical power with this sample size. It is important to show through power calculation that the designed sample size can give your analysis sufficient power.

RESPONSE: We have not powered this study in the way one might for an efficacy trial. We have proposed a sample size that enables an assessment of the main uncertainties for a future definitive trial.

3. Clustering issue: the authors mentioned on p.2, line 23-29 "48 participants... will be recruited from three national health service hospitals...". Therefore, it is helpful to clarify the following questions: Does it mean there will be clusters in the recruitment? Does this need to be addressed in the statistical analysis? How is randomization conducted within/between hospitals? More details/clarifications are necessary.

RESPONSE: Clustering at centres and related to therapists are very valid concerns for a rehabilitation trial. While we do report our stratification of randomisation by centre for this pilot study, we are not conducting a definitive trial, or analysing using formal statistical tests of treatment effect. We appreciate that these will be important considerations in the design and analysis for a definitive trial if this pilot trial indicates this would be feasible to conduct.

4. Mediating effect of psychological barrier: on p.6. 28-32, the authors mentioned that “psychological barriers to adherence to physiological advice and exercises... will be addressed”. Can the authors provide more details on how this issue could be addressed in your trial? Do people under the two treatments experience different psychological barriers in average? How would this be addressed/balanced in the design/analysis?

RESPONSE: The psychological barriers to adherence to exercise and advice should be balanced between intervention groups by randomisation. We report that we will be assessing self-efficacy in both groups. The progressive exercise intervention description is the part of the manuscript that outlines how these are addressed by the progressive exercise intervention:

‘The progressive functional exercise intervention will use simple health behaviour change techniques to optimise adherence to home exercise. We drew on the evidence-based NHS Health Trainers Handbook, recommended for routine use by health professionals to promote patient behaviour change.³² Our group has used these techniques in other physiotherapy trials to develop feasible exercise programmes that have resulted in self-efficacy improvements.³³ The strategies use a twostage mechanism, increasing intention to adhere to the exercise regimen and translating this behavioural intention into actual behaviour.

Participants will be asked to identify their goals following usual physiotherapy practice and, with the treating therapist’s help, write an action plan for where and when they will perform their home exercises and a contingency plan for managing difficulties.

Therapists will be trained to focus on helping participants identify barriers to exercise and becoming more physically active post-injury, and facilitating problem-solving. The therapists will offer education on how exercise and physical activity can help participants to achieve their goals and will reassure participants about their capacity to exercise and increase their physical activity.³⁴ The intervention will give participants individualised feedback on their rehabilitation progress and reinforcement over the sessions, and will facilitate identification of barriers to doing the home exercise programme, which all have a strong evidence base to support their use in older adults.²⁹’

5. The statistical analysis section on p.15-16 needs to be improved. The current analytical plan seems to only calculate the group means and standard deviations. Is there any power calculation for this analysis? Should more information/covariates be included in the analysis since there are only 24 participants in each arm? Personal thoughts: at least, some additional variate adjustment analysis is necessary to improve the power. Also, because the participants are not blinded and the noncompliance rate might be high, the ITT (intention to treatment) estimand (true effect size) could naturally be very small. It is therefore even more important to show whether 24 people in each treatment arm can provide enough power to your analysis.

RESPONSE: This comment appears to suggest we will be calculating treatment effects, whereas this is a feasibility study that is not designed to assess these. We are reporting feasibility outcomes and variability in outcome measures in order to ascertain that a fully powered definitive trial would be feasible to be conducted. We will therefore not be conducting definitive analyses or adjusted estimates of effects. These would form part of a fully powered definitive trial if it proves feasible.

6. Missing data: it is likely that the current trial will encounter missing data/dropout issues. It is discussed in the feasibility section (p.12-13). However, how the missing data would be addressed in

the analysis, such as by using weighting- or imputation-based methods, is not discussed. The validity of the analysis is likely to be based on the “missing at random” assumption. And if there is any informative dropout, such as psychological barrier/lack of efficacy, how can this information be incorporated into the analysis? Some more discussion is needed for this.

RESPONSE: As above, these comments indicate that our study is focussing on a definitive assessment of treatment effect, which is not the aim of the study. We will however be reporting missingness and loss to follow-up as these will be important parameters to aid design of a later definitive trial.

7. Embedded qualitative study: it is not particularly clear to me whether the embedded qualitative study is directly related to this trial design. While the authors state on p.16, “...will give insights into the acceptability of the trial interventions and randomization to them,” the analysis of this trial will be based on ITT (intention to treatment), which means the data collected in this embedded qualitative study will not affect the conclusion of the study. The authors need to either provide a stronger justification over this part or remove it.

RESPONSE: Thank you for your comment. The study is a feasibility study rather than the main trial so the following has been added into the text to provide clarity on this point. Inserted:

'The embedded qualitative study aims to find out more about the patients' experience of the two interventions within the context of their recovery from ankle fracture. This understanding will help us review the acceptability of the two interventions, which aspects help or hinder recovery. This will enable us to refine the interventions, retaining aspects that are important to patients and developing or removing those that are less helpful. For instance, patients may struggle with the pace or complexity of the progressive exercises, which could then be modified for the future definitive trial. In addition the interviews will provide valuable insight into how patients experience the trial processes. For example we know that patients take part in studies because they place their trust in the clinical team, they can have trouble understanding some information, can find randomisation unacceptable in clinical situations and have therapeutic misconceptions.⁴⁸⁻⁵⁰ Gaining an insight into the aspects that facilitate and limit participation in the study will help us refine our information processes to include areas that are of concern to patients.'

8. Minor point: I am confused about the column names in Table 1, on p.24, why are the sessions appeared in 0-4 months, but the first follow up is 3-month? So, it is possible for data collection/followup to happen even before the treatment sessions are finished? Some clarifications are needed here.

RESPONSE: Thank you for highlighting this. The three month follow-up will occur while participants are still either working on their self-management or seeing their physiotherapist, there is a later 6 month follow up, which is afterward. We have added a footnote to clarify.

VERSION 2 – REVIEW

REVIEWER	Guangyu Tong Duke University School of Medicine
REVIEW RETURNED	19-Sep-2019

GENERAL COMMENTS	The authors provided thoughtful responses to my questions.
--